# Effect of Astaxanthin on the Inhibition of Lipid Accumulation in 3T3-L1 Adipocytes via Modulation of Lipogenesis and Fatty Acid Transport Pathways

**DOI:** 10.3390/molecules25163598

**Published:** 2020-08-07

**Authors:** Mei-Chih Tsai, Shih-Chien Huang, Wei-Tang Chang, Shiuan-Chih Chen, Chin-Lin Hsu

**Affiliations:** 1Department of Nutrition, Chung Shan Medical University, Taichung 40201, Taiwan; drmaginutrigene@yahoo.com (M.-C.T.); schuang@csmu.edu.tw (S.-C.H.); 2Department of Nutrition and Health Nutrition, Chinese Culture University, Taipei 11114, Taiwan; zwt6@ulive.pccu.edu.tw; 3Institute of Medicine and School of Medicine, Chung Shan Medical University, Taichung 40201, Taiwan; sccy399@gmail.com; 4Department of Family and Community Medicine, Chung Shan Medical University Hospital, Taichung 40201, Taiwan; 5Department of Nutrition, Chung Shan Medical University Hospital, Taichung 40201, Taiwan

**Keywords:** astaxanthin, 3T3-L1 adipocytes, lipogenesis, gene expression

## Abstract

Obesity is defined as a condition of excessive fat tissue accumulation. It was the major factor most closely associated with lifestyle-related diseases. In the present study, we investigated the effect of astaxanthin on the inhibition of lipid accumulation in 3T3-L1 adipocytes. 3T3-L1 adipocytes were treated with 0–25 µg/mL of astaxanthin for 0–48 h. The result indicated that astaxanthin significantly decreased the oil Red O stained material (OROSM), intracellular triglyceride accumulation, and glycerol 3-phosphate dehydrogenase (GPDH) activity in 3T3-L1 adipocytes (*p* < 0.05). At the molecular level, astaxanthin significantly down-regulated the mRNA expression of *peroxisome proliferator-activated receptor-γ* (*PPARγ*) in 3T3-L1 adipocytes (*p* < 0.05). Moreover, target genes of *PPARγ* on the inhibition of lipogenesis, such as *Acetyl-CoA carboxylase* (*ACC*), *fatty acid synthase* (*FAS*), *fatty acid binding protein* (*aP2*), *cluster of differentiation 36* (*CD36*), and *lipoprotein lipase* (*LPL*) in 3T3-L1 adipocytes were significantly down-regulated at a time-dependent manner (*p* < 0.05). These results suggested that astaxanthin efficiently suppressed lipid accumulation in 3T3-L1 adipocytes and its action is associated with the down-regulation of lipogenesis-related genes and the triglyceride accumulation in 3T3-L1 adipocytes. Therefore, astaxanthin can be developed as a potential nutraceutical ingredient for the prevention of obesity in a niche market.

## 1. Introduction

The American Medical Association (AMA) officially recognized obesity as a chronic disease in 2008 [1]. It was a significant risk factor for various diseases such as type 2 diabetes, dyslipidemia, cancer, coronary heart disease, and hypertension [2]. Obesity is a state in which a large amount of lipid droplets accumulate, thus differentiating into a mature adipocyte [3]. The evidence suggests that visceral adipose tissue (VAT) is a highly active metabolic and endocrine organ that secretes several inflammatory cytokines, which influences not only the body-weight homeostasis, but also the metabolism of the liver, muscle, and cardiovascular systems [4]. Glycerol-3-phosphate dehydrogenase (GPDH) is an enzyme that appears to have an important role in the conversion of glycerol into triglyceride [5]. Many studies have reported a close relationship between obesity and oxidative stress [6,7]. Oxidative stress could play a significant role in the development of obesity by stimulating white adipose tissue deposition. Oxidative stress is associated with increased preadipocyte proliferation, adipocyte differentiation, and the size of mature adipocytes in vitro and in vivo [8,9,10].

Many natural dietary compounds (such as flavonoids, stilbene, diarylheptanoids, carotenoids, and phenolics) have exhibited several beneficial effects for the treatment of diseases [11]. Astaxanthin is a class of natural colorful pigments present in marine animals, which have a natural antioxidant and fat-soluble molecule. It is naturally found most abundantly in *Haematococcus pluvialis*, which accumulates high astaxanthin content under high stress conditions [12]. Olaizola et al. (2008) [13] indicated that astaxanthin possesses antioxidant activity several folds higher than those of other carotenoids and tocopherols. Recently, astaxanthin studies have focused on several biological functions, such as radical scavenging, singlet oxygen quenching, anti-carcinogenesis, anti-diabetic, anti-obesity, anti-inflammatory, and immune enhancement activities. Moreover, the outstanding antioxidant properties of astaxanthin are considered to contribute to the protection of organisms against obesity [14]. Furthermore, astaxanthin enhanced systemic fatty acid utilization, as indicated by reduced respiratory quotient in indirect calorimetry tests [15]. It can also inhibit endoplasmic reticulum stress and lipogenesis at the intracellular level [16]. Peroxisome proliferator-activated receptor (PPAR) has been found to be an important modulator of adipogenesis. PPARγ coordinately drives the expression of adipocyte-specific genes such as adipocyte fatty acid-binding protein 2 (aP2), fatty acid synthase (FAS), acetyl-CoA carboxylase (ACC), lipoprotein lipase (LPL), and cluster of differentiation 36 (CD 36), many of which characterize the final stages of differentiation [17]. Astaxanthin is a PPARγ antagonistic effect on adipocytes [18,19]. However, the literature regarding the effect of astaxanthin on the inhibition of lipid accumulation in 3T3-L1 adipocytes and the molecular mechanisms involved in the regulation of lipid accumulation remains unclear.

## 2. Results

### 2.1. Effect of Astaxanthin on Oil Red O Stained Material (OROSM) in 3T3-L1 Adipocytes

Figure 1 and Figure 2 show the effect of astaxanthin on oil red stained material (OROSM) in 3T3-L1 adipocytes. Cell number analysis of 3T3-L1 was performed by OROSM after being treated with 0, 1, 5, 10, and 25 µg/mL astaxanthin for 24 and 48 h. As shown in Figure 1 and Figure 2, OROSM indicated that the astaxanthin (treated with 25 µg/mL for 24 and 48 h) had significantly decreased adipocyte numbers to 94.56% (24 h) and 86.21% (48 h), respectively (*p* < 0.05). Moreover, treatment with 1–10 µg/mL of astaxanthin showed no significant reduction in the level of OROSM at 24 and 48 h in 3T3-L1 adipocytes.

### 2.2. Effect of Astaxanthin on the Content of Intracellular Triglyceride in 3T3-L1 Adipocytes

The effect of astaxanthin on the inhibition of intracellular triglyceride in 3T3-L1 adipocytes is shown in Figure 3. The results demonstrated that astaxanthin caused an inhibition of intracellular triglyceride in the 3T3-L1 adipocytes. The data revealed that the astaxanthin (treated with 25 µg/mL for 24 h) significantly decreased the level of intracellular triglyceride in 3T3-L1 adipocytes (*p* < 0.05) (from 100% to 89.10%). Moreover, astaxanthin (1–25 µg/mL, 48 h) also significantly decreased the level of intracellular triglyceride in 3T3-L1 adipocytes (*p* < 0.05) to 79.24%, 65.59%, 80.08%, and 76.88%, respectively.

### 2.3. Effect of Astaxanthin on GPDH Activity in 3T3-L1 Adipocytes

The effect of astaxanthin on GPDH activity in 3T3-L1 adipocytes is shown in Figure 4. The results demonstrated that astaxanthin caused the inhibition of GPDH activity in 3T3-L1 adipocytes. Our data indicated that the astaxanthin (5, 10, and 25 µg/mL for 48 h) significantly decreased the GPDH activity in 3T3-L1 adipocytes (81.62%, 81.73%, and 77.79%, respectively) (*p* < 0.05).

### 2.4. Effect of Astaxanthin on Gene Expression Levels of Fatty Acid Synthesis in 3T3-L1 Adipocytes

Figure 5 shows the effect of astaxanthin on fatty acid synthesis-related gene expressions levels of *PPARγ*, *ACC*, and *FAS* in 3T3-L1 adipocytes. Astaxanthin (25 µg/mL, 3–9 h) remarkably decreased the gene expression of *PPARγ* in 3T3-L1 adipocytes. The data also indicated that astaxanthin (25 µg/mL, 9 h) significantly decreased the gene expressions of *ACC* and *FAS* in 3T3-L1 adipocytes (*p* < 0.05).

### 2.5. Effect of Astaxanthin on Gene Expression Levels of Fatty Acid Transport in 3T3-L1 Adipocytes

Figure 5 shows the effect of astaxanthin on fatty acid transport-related gene expression levels of *aP2*, *CD36*, and *LPL* in 3T3-L1 adipocytes. Astaxanthin (25 µg/mL, 9 h) remarkably decreased the gene expression of aP2 in 3T3-L1 adipocytes. The data also indicated that Astaxanthin (25 µg/mL, 6–9 h) significantly decreased the gene expressions of *CD36* and *LPL* in 3T3-L1 adipocytes (*p* < 0.05).

## 3. Discussion

Obesity has become a known globalized health issue. Accompanied by many negative effects on the human body, it is also a major contributing factor to many diseases. It is the result of energy intake exceeding energy expenditure over an extended period. [20]. Wang and Jones indicated that the decreased adipocytic lipogenesis is one of the mechanisms of proposed anti-obesity [21].

Astaxanthin is the most common carotenoid in marine organisms, such as algae and aquatic animals. Previous studies indicated that astaxanthin has multiple pharmacological properties including antioxidant, anti-inflammatory, anti-cancer activities, and anti-adiposity action [22]. Therefore, adipose tissue mass represents an attractive concept for combating obesity and associated metabolic diseases [23]. In the present study, we focused on the effect of astaxanthin on the inhibition of lipid accumulation in 3T3-L1 adipocytes. This inhibitory effect resulted from the repression of adipocyte-specific gene expressions. The OROSM was expressed relative to the numbers of adipocytes counted on comparable plates, decreased to 94.56% at 24 h and to 86.21% at 48 h in astaxanthin treated 3T3-L1 adipocytes (Figure 1 and Figure 2). Inoue et al. [18] demonstrated that the astaxanthin (10 μM) significantly reduced lipid accumulation of 3T3-L1 adipocytes by specifically inhibiting PPARγ transcriptional activity. Jia et al. [24] also showed that astaxanthin significantly reduced intracellular triglyceride and cholesterol when HepG2 cells were treated with 5 and 10 μM astaxanthin for 24 h.

GPDH is the predominant substrate for triglyceride synthesis in adipose tissue [25]. The results showed that exposure of 3T3-L1 adipocytes to astaxanthin caused the inhibition on intracellular triglyceride and GPDH activity in a dose-dependent manner (Figure 3 and Figure 4). Previously reports have proposed several strategies to improve lipid metabolism [21]. Price et al. (2012) [26] indicated the differences signaling pathways resveratrol treatment have at different doses. Therefore, astaxanthin could be a valuable component to improve obesity in optimal doses (5 µg/mL). PPARγ regulates fatty acid storage via activation of genes that stimulate lipid uptake and adipogenesis by fat cells [27]. Thus, overexpression of PPARγ induces lipid accumulation in adipose tissue. For gene expressions, astaxanthin significantly down-regulated the mRNA expression levels of PPARγ and lipogenesis genes (ACC and FAS). Our previous studies showed that astaxanthin inhibited the expressions of lipogenesis at mRNA levels in 3T3-L1 adipocytes (Figure 5). Zhao et al. [28] reported that carotenoids (such as bixin, lycopene, and β-carotene) significantly decreased lipid accumulation in 3T3-L1 adipocytes and markedly down-regulate the protein levels of PPARγ, FABP4, leptin, and ACC.

Activation of PPARγ by its ligands alone also leads to adipocyte differentiation and aP2 expression [29]. Adipocyte fatty acid binding protein (aP2) is a key mediator of intracellular transport and metabolism of fatty acids. Expression of peroxisome proliferator-activated receptor gamma (PPARγ) and CCAAT/enhancer binding protein alpha (C/EBPα) promotes induction of the adipocyte fatty acid binding protein (aP2) in 3T3-L1 adipocytes [30].

FAT/CD36 has an important role for long-chain fatty acids (LCFAs) as a transporter to facilitate the uptake of LCFAs in adipocytes [31]. CD36 expression is up-regulated by PPARγ during the differentiation of adipocyte and macrophage [32]. Our results show that astaxanthin significantly down-regulated the mRNA expression levels of *adipocyte lipid-binding protein 2 (aP2)* and *cluster of differentiation 36 (CD36)* (Figure 5).

Lipoprotein lipase (LPL) is the enzyme for the import of triglyceride-derived fatty acids by muscle for utilization and adipose tissue for storage [33]. Overexpression of lipoprotein lipase (LPL) led to profound localized lipid in endothelialized artery [34]. These results show that astaxanthin also down-regulated the mRNA expression of *LPL* (Figure 5).

## 4. Experimental Section

### 4.1. Materials

Astaxanthin (LemnaRed^®^ astaxanthin crystal is manufactured via a fermentation process) was purchased from Lemnaceae Fermentation, Inc. (Taoyuan City, Taiwan). The crystal was compared with Sigma astaxanthin product and identified using high performance liquid chromatography (HPLC) chromatogram (Shimadzu Corporation, Kyoto, Japan) (Figure 6A,B). The wavelength of maximum absorbance is at 477 ± 3 nm and identified with standard product (Figure 6C) (Merck KGaA, Darmstadt, Germany). The total carotenoids content is approximately 93.3%. Total carotenoids contain 91.5% astaxanthin, 3.4% adonirubin, and 3.9% canthaxanthin. It is also worth noting that after the extraction process there is no caller DNA or protein remaining in the LemnaRed^®^ astaxanthin crystal. In addition, astaxanthin in the LemnaRed^®^ astaxanthin crystal is not esterified by fatty acids, and contains 86.3% (3S, 3S’)—all trans astaxanthin, 3.4% 13-cis astaxanthin, and 1.7% 9-cis astaxanthin (Figure 6D).

### 4.2. 3T3-L1 Cells Differentiation

3T3-L1 preadipocytes (BCRC 60159, cell passage number is 10) were purchased from the Bioresource Collection and Research Center (BCRC, Food Industry Research and Development Institute, Hsinchu, Taiwan). 3T3-L1 preadipocytes were planted into 6-well plates and maintained in DMEM supplemented with 10% bovine calf serum at 37 °C in a humidified 5% CO_2_ incubator. 3T3-L1 preadipocytes differentiation was induced by the adipogenic agents (0.5 mM 3-isobutyl-1-methylxanthine, 1 µM dexamethasone, and 1 µM insulin) and added to a culture medium. On day 8, differentiation media was removed from the 3T3-L1 adipocytes and normal culture medium was added, and freshly replaced every 48 h.

### 4.3. Oil Red O Staining

Intracellular lipid accumulation was measured using oil red O staining. The oil red O working solution was prepared as described by Ramirez Zacarias et al. [35]. After ten days of differentiation, the 3T3-L1 adipocytes culture to become mature 3T3-L1 adipocytes. Cells were incubated with 0, 1, 5, 10, and 25 µg/mL of astaxanthin for 24 and 48 h at 37 °C in a humidified 5% CO_2_ incubator. Cells were washed twice with phosphate buffered saline (PBS, pH 7.4) and then fixed with 10% neutral formalin for 20 min at room temperature. After the 10% neutral formalin was removed, 100% propylene glycol was added to each well for 3 min. Cells were decolorized with 60% propylene glycol before staining for 1 h with the oil red O working solution and then dried at 37 °C in an oven for 1 h. The staining dye of cells was extracted with isopropyl alcohol (1 mL/well) and measured spectrophotometrically at 510 nm in an ASYS UVM 340 microplate reader (Biochrom, Holliston, MA, USA). The OROSM was expressed on a per cell basis using the cell number determined from similar plates. The percentage of oil red O staining material (OROSM, %) relative to control wells containing cell culture medium without compounds was calculated as A510 nm [antioxidant]/A510 nm [control] × 100.

### 4.4. Intracellular Triglyceride Accumulation

After ten days of differentiation, the 3T3-L1 adipocytes culture to become mature 3T3-L1 adipocytes. Cells were incubated with 0, 1, 5, 10, and 25 µg/mL of astaxanthin for 24 and 48 h at 37 °C in a humidified 5% CO_2_ incubator. Cells were collected and lysed in lysis buffer (1% Triton X-100 in PBS). The content of total intracellular triglyceride in 3T3-L1 adipocytes was determined using a commercial triglyceride assay kit (DiaSys Diagnostic Systems GmbH, Holzheim, Germany). The content of total protein was determined by using a BioRad DC protein assay kit (Bio-Rad Laboratories, Hercules, CA, USA). Inhibition (%) was expressed as percent decrease in triglyceride content against control (0%).

### 4.5. Glycerol-3-phosphate Dehydrogenase (GPDH) Activity

After ten days of differentiation, the 3T3-L1 adipocytes culture to become mature 3T3-L1 adipocytes. Cells were incubated with 0, 1, 5, 10, and 25 µg/mL of astaxanthin for 24 and 48 h at 37 °C in a humidified 5% CO_2_ incubator. Cells were washed twice carefully with ice-cold PBS, and lysed in 25 mM Tris/1 mM EDTA (pH 7.5) for the measurement of glycerol-3-phosphate dehydrogenase (GPDH) specific activity. GPDH activity was determined according to the procedure of Wise and Green [5]. The content of total protein was determined by the BioRad DC protein assay kit (Bio-RadLaboratories, Hercules, CA, USA), which uses bovine serum albumin as a standard. Enzyme activity was expressed as units of activity/mg protein. GPDH activity is expressed as a percentage, with the value of control set at 100%.

### 4.6. RNA Extraction and Real-Time Reverse Transcription-Polymerase Chain Reaction

Total RNA was isolated from the 3T3-L1 cells using Trizol reagent (Invitrogen, Carlsbad, CA, USA) following the manufacture’s recommendations. The following primer pairs in 3T3-L1 adipocytes were used: GAPDH, 5′-gtatgactccactcacggcaaa-3′ (forward) and 5′-ggtctcgctcctggaagatg-3′ (reverse); PPARγ, 5′-ttttcaagggtgccagtttcgatcc-3′ (forward) and 5′-aatccttggccctctgagat-3′ (reverse); FAS, 5′-tgggttctagccagcagagt-3′ (forward) and 5′-taccaccagagaccgttatgc-3′ (reverse); ACC, 5′-gaatctcctggtgacaatgcttatt-3′ (forward) and 5′-ggtcttgctgagttgggttagct-3′ (reverse); aP2, 5′-catggccaagcccaacat-3′ (forward) and 5′-cgcccagtttgaaggaaatc-3′ (reverse); CD36, 5′-gcttgcaactgtcagcacat-3′ (forward) and 5′-gccttgctgtagccaagaac-3′ (reverse); LPL, 5′-atcggagaactgctcatgatga-3′ (forward) and 5′-cggatcctctcgatgacgaa-3′ (reverse). Real-time quantitative polymerase chain reaction (PCR) was performed with an SYBR Green^TM^ kit (Quantitect^TM^ SYBR Green PCR; QIAGEN, Valencia, CA, USA). The cycling conditions were 15 min at 95 °C, 40 cycles of 15 s at 94 °C, 30 s at 51 °C, and 30 s at 72 °C. Relative quantification was performed using the ΔΔCt method [36].

### 4.7. Statistical Analysis

All of the data are expressed as means ± standard deviation. The data were analyzed using analysis of variance (ANOVA). Differences between groups were assessed using Duncan’s test. All statistical analyses of the data were performed using the SPSS software version 13.0 (SPSS Inc., Chicago, IL, USA). A *p* value of <0.05 was considered statistically significant.

## 5. Conclusions

This study indicates that astaxanthin could inhibit lipogenesis and lipid deposition in 3T3-L1 adipocytes. Our results indicate that astaxanthin could decrease the intracellular triglyceride, OROSM, and GPDH activity in 3T3-L1 adipocytes. Astaxanthin suppresses the gene expressions levels of lipogenesis (*PPARγ*, *FAS*, and *ACC*) and fatty acid transport (*aP2*, *CD36*, and *LPL*) in 3T3-L1 adipocytes. These findings suggest that astaxanthin have a potent antioxidant carotenoid and may provide a potential therapeutic approach for the treatment of obesity. However, further studies on the molecular action mechanisms as well as clinical investigations are strongly recommended.

## Figures and Tables

**Figure 1 molecules-25-03598-f001:**
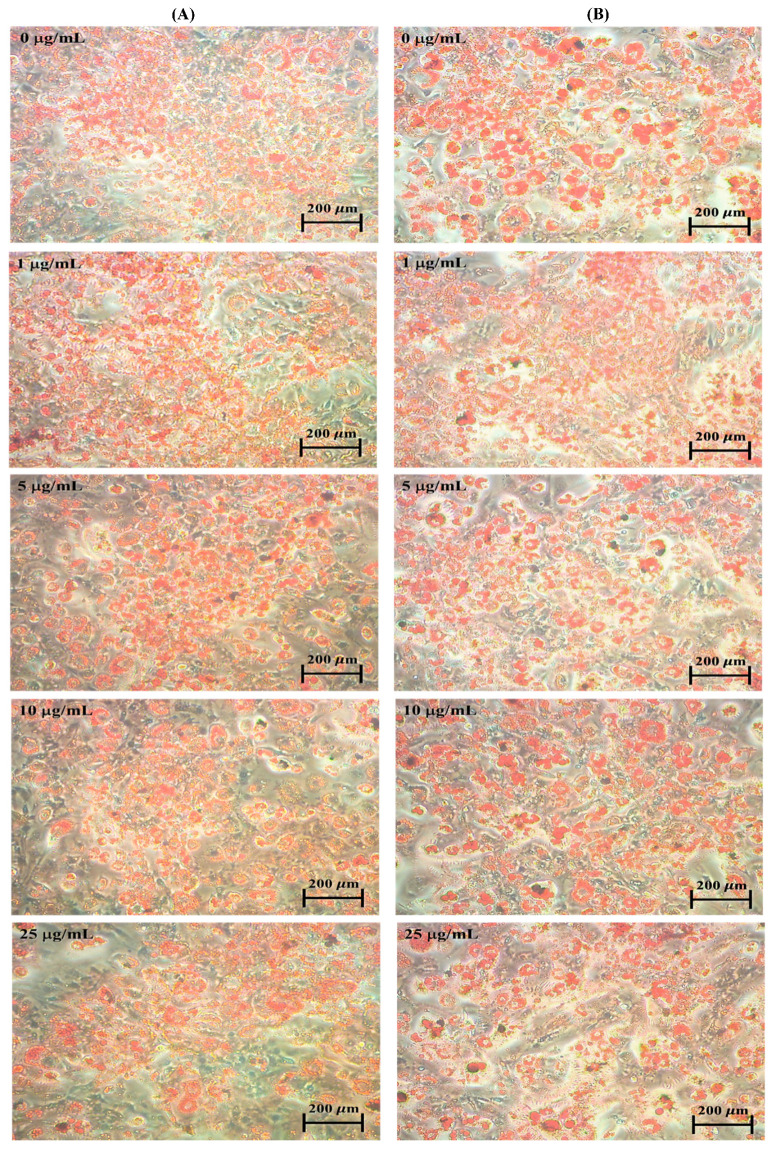
Effect of astaxanthin on cell morphological in 3T3-L1 adipocytes. 3T3-L1 adipocytes were stained with oil red O stained material (OROSM) after treatment with 0–25 μg/mL of astaxanthin for 24 h (**A**) and 48 h (**B**) at 37 °C in a 5% CO_2_ incubator. Original magnification: 200×.

**Figure 2 molecules-25-03598-f002:**
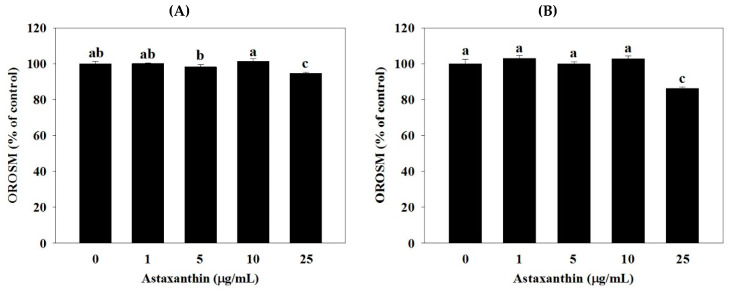
Effect of astaxanthin on OROSM in 3T3-L1 adipocytes. Percentage (%) is expressed as control at 100%. 3T3-L1 adipocytes were incubated with 0–25 μg/mL of astaxanthin for 24 h (**A**) and 48 h (**B**) at 37 °C in 5% CO_2_ incubator. The reported values are the means ± SD (n = 3). Mean values with different letters were significantly different (*p* < 0.05).

**Figure 3 molecules-25-03598-f003:**
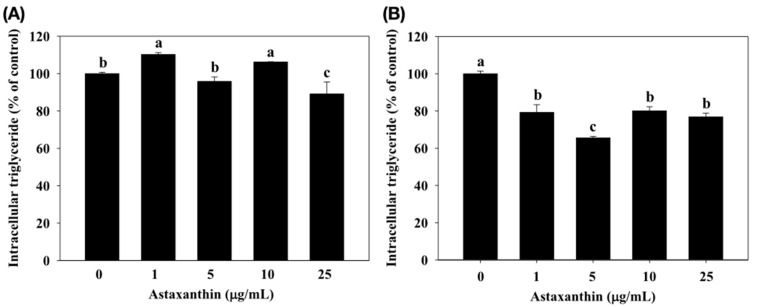
Effect of astaxanthin on inhibition of intracellular triglyceride in 3T3-L1 adipocytes. Percentage (%) is expressed as control at 100%. 3T3-L1 adipocytes were incubated with 0–25 μg/mL of astaxanthin for 24 h (**A**) and 48 h (**B**) at 37 °C in 5% CO_2_ incubator. The reported values are the means ± SD (n = 3). Mean values with different letters were significantly different (*p* < 0.05).

**Figure 4 molecules-25-03598-f004:**
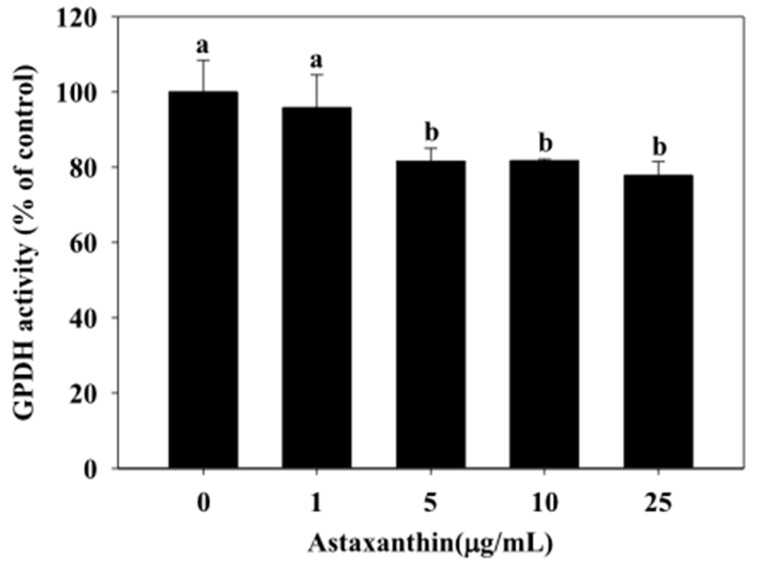
Effect of astaxanthin on glycerol-3-phosphate dehydrogenase (GPDH) activity in 3T3-L1 adipocytes. Percentage (%) is expressed as control at 100%. 3T3-L1 adipocytes were incubated with 0–25 μg/mL of astaxanthin for 48 h at 37 °C in 5% CO_2_ incubator. The reported values are the means ± SD (n = 3). Mean values with different letters were significantly different (*p* < 0.05).

**Figure 5 molecules-25-03598-f005:**
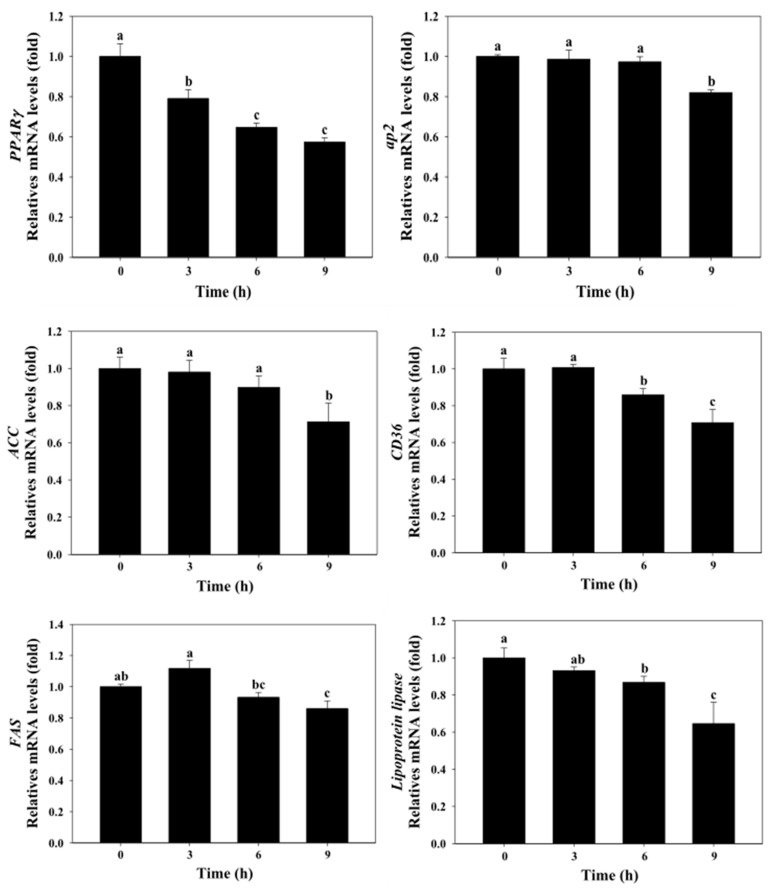
Effect of astaxanthin on gene expression levels of *PPARγ*, *ACC*, *FAS*, *aP2*, *CD36*, and *LPL* in 3T3-L1 adipocytes. The reported values are the means ± SD (n = 3). Mean values with different letters were significantly different (*p* < 0.05). 3T3-L1 adipocytes were treated with 25 μg/mL of astaxanthin for 0, 3, 6, and 9 h at 37 °C in a humidified 5% CO_2_ incubator. The values of *PPARγ*, *ACC*, *FAS*, *aP2*, *CD36*, and *LPL* mRNA were normalized to the value of *GAPDH*.

**Figure 6 molecules-25-03598-f006:**
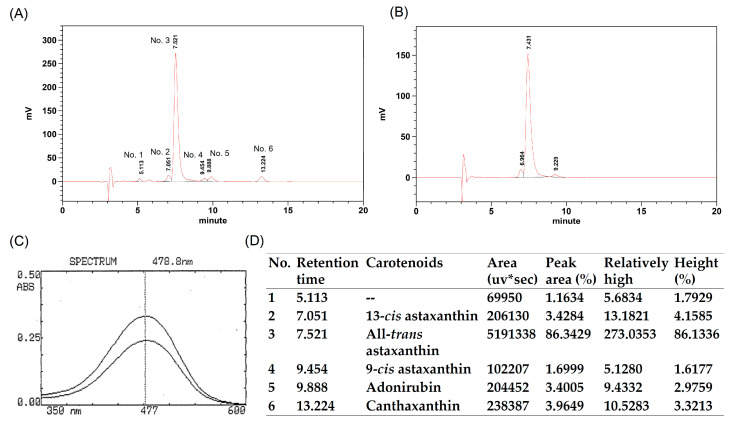
High performance liquid chromatography (HPLC) chromatogram of astaxanthin (**A**) and standard (**B**). The absorption spectrum of astaxanthin from LemnaRed and standard using UV-VIS Spectrophotometer (**C**). Total carotenoids content of LemnaRed sample (**D**).

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
