# Peer review of "Effect of Astaxanthin on the Inhibition of Lipid Accumulation in 3T3-L1 Adipocytes via Modulation of Lipogenesis and Fatty Acid Transport Pathways"

_molecules, 2020, doi:10.3390/molecules25163598_

Round 1

Reviewer 1 Report

The authors of the manuscript titled “ Effect of astaxanthin on the inhibition of adipogenesis in 3T3-L1 adipocytes via modulation of lipogenesis and fatty acid transport pathways” wrote in abstract “we investigated that the effect of astaxanthin on the inhibition of adipogenesis in 3T3-L1 adipocytes”. In my opinion, it is not true, the results didn’t show such effect, and the effect of astaxanthin was evaluated on mature adipocytes after completion of the differentiation process. The title is wrong and misleading!

Moreover, this paper contains meaningful issues that need to be explained. First of all, the language and sentence construction should be checked. The conditions of some experiments are lacked and not clear. The methods and results have to be carefully and clearly explained. The most critical issues are listed below:
1. What is a difference and novelty in this paper in comparison with study Inoue, M.; Tanabe, H.; Matsumoto, A.; Takagi, M.; Umegaki, K.; Amagaya, S.; Takahashi, J. Astaxanthin functions differently as a selective peroxisome proliferator-activated receptor g modulator in adipocytes and macrophages. Biochem. Pharmacol. 2012, 84, 692-700 in which authors have shown that astaxanthin inhibited adipogenesis in 3T3-L1 cells? which showed that astaxanthin completely inhibited rosiglitazone-induced lipid accumulation and reduced the mRNA expression of PPARγ target genes (aP2, fatty acid-binding protein, and LPL) in the 3T3-L1 adipocyte.
2. The 3T3-L1 cell line was not purchased from ATCC or ETCC, which are accredited organizations for selling authentic cell lines. In this study were used cells from Bioresource Collection and Research Center (BCRC, Food Industry Research and Development Institute, Hsinchu, Taiwan) did authors checked the authentication of this cell line? In which passage number was initially this cell line?
3. Incomplete description of differentiation of adipocytes, which day was adipogenic agents added to the cells, how was confluency of cells when the adipogenic agents were added, what was the next step after adding IBMX, DEX and insulin?
4. Line 191- what it means that 3T3-L1 adipocytes were harvested ?? If it means that adipocytes were detached and seeded into new plates?
5. Astaxanthin was added to cell culture ten days after the differentiation process so the effect of astaxanthin was evaluated on mature adipocytes, not on adipogenesis!
6. There is no cytotoxicity-viability test of astaxanthin on adipocytes; maybe this lipid-lowering effect was due to cytotoxicity of astaxanthin?
7. The real-time PCR was normalized to GAPDH, not to b-actin, GAPDH expression levels increased during the course of 3T3-L1 adipocyte differentiation and b-actin show relatively stable expression level [Zhang J, Tang H, Zhang Y, Deng R, Shao L, Liu Y, Li F, Wang X, Zhou L. Identification of suitable reference genes for quantitative RT-PCR during 3T3-L1 adipocyte differentiation. Int J Mol Med. 2014 May;33(5):1209-18. doi: 10.3892/ijmm.2014.1695]. In this study have been found that the mRNA expression levels of glyceraldehyde-3-phosphate dehydrogenase (GAPDH) and transferrin receptor (TFRC)significantly increased during 3T3-L1 adipocyte differentiation.
8. In my opinion, this is a preliminary study which examined only one aspect – the effect of astaxanthin on lipid accumulation in mature adipocytes, which was examined by Oil Red and triglyceride assay kit and six gene expression, so the sentence in conclusions “These findings suggested that astaxanthin have a potent antioxidant carotenoid, as a promising natural candidate for prevented lifestyle diseases and health promotion” is overstated and more study is needed to be done to drawing such conclusions (like investigation on lipolysis, adipogenesis, leptin, adiponectin expression etc.)
9. Fig. 2, the photos of adipocytes are not sharp and not clear; it is impossible to see round cells filled with lipids droplets. Moreover, all photos should be taken in the same zoom that is possible to see adipocytes morphology, here adipocytes seem to be deformed; it looks like there is a cytotoxic effect of astaxanthin.

Author Response

Reviewer 1

Comments and Suggestions for Authors

The authors of the manuscript titled “ Effect of astaxanthin on the inhibition of adipogenesis in 3T3-L1 adipocytes via modulation of lipogenesis and fatty acid transport pathways” wrote in abstract “we investigated that the effect of astaxanthin on the inhibition of adipogenesis in 3T3-L1 adipocytes”. In my opinion, it is not true, the results didn’t show such effect, and the effect of astaxanthin was evaluated on mature adipocytes after completion of the differentiation process. The title is wrong and misleading!

Moreover, this paper contains meaningful issues that need to be explained. First of all, the language and sentence construction should be checked. The conditions of some experiments are lacked and not clear. The methods and results have to be carefully and clearly explained. The most critical issues are listed below:

  1. What is a difference and novelty in this paper in comparison with study Inoue, M.; Tanabe, H.; Matsumoto, A.; Takagi, M.; Umegaki, K.; Amagaya, S.; Takahashi, J. Astaxanthin functions differently as a selective peroxisome proliferator-activated receptor g modulator in adipocytes and macrophages. Biochem. Pharmacol. 2012, 84, 692-700 in which authors have shown that astaxanthin inhibited adipogenesis in 3T3-L1 cells? which showed that astaxanthin completely inhibited rosiglitazone-induced lipid accumulation and reduced the mRNA expression of PPARγ target genes (aP2, fatty acid-binding protein, and LPL) in the 3T3-L1 adipocyte.

RESPONSE: We agreed the reviewer’s opinion. The difference between our study and Inoue et al. (2012) are the cell culture method. Inoue et al. (2012) mentioned that astaxanthin (ASX) or rosiglitazone (RGZ) (PPARg agonist) was added to the post-differentiation medium, and incubated for 4 days. In the present study, astaxanthin was added to differentiated mature adipocytes for 24 and 48 hours (without rosiglitazone). Obviously, the purpose of this study was to investigated that the effect of astaxanthin on lipid accumulation in mature adipocytes. However, the research of Inoue et al. (2012) is mainly to explore the differentiation of 3T3-L1 adipocytes. Even if the conditions for astaxanthin to treat adipocytes are different, the results of both studies have shown that astaxanthin could inhibit the PPAR gamma signal pathway. It can be used as a useful information for the future discussion of astaxanthin to reduce fat accumulation.

  1. The 3T3-L1 cell line was not purchased from ATCC or ETCC, which are accredited organizations for selling authentic cell lines. In this study were used cells from Bioresource Collection and Research Center (BCRC, Food Industry Research and Development Institute, Hsinchu, Taiwan) did authors checked the authentication of this cell line? In which passage number was initially this cell line?

RESPONSE: We acknowledge that we did not checked the authentication of this cell line. In addition, the procedure for purchasing cell lines from ATCC is relatively complicated in Taiwan, so the 3T3-L1 preadipocytes cell line was purchased from Bioresource Collection and Research Center (BCRC, Food Industry Research and Development Institute, Hsinchu, Taiwan). BCRC number is 60159 and the cell passage number is 10. (p. 8, line 186)

  1. Incomplete description of differentiation of adipocytes, which day was adipogenic agents added to the cells, how was confluency of cells when the adipogenic agents were added, what was the next step after adding IBMX, DEX and insulin?

RESPONSE: Thank you for your suggestion. We have rewritten 3T3-L1 cells differentiation protocol (line 189-193) to be more in line with your comments.

  1. Line 191- what it means that 3T3-L1 adipocytes were harvested ?? If it means that adipocytes were detached and seeded into new plates?

RESPONSE: Thank you for your suggestion. In the article “3T3-L1 adipocytes were harvested” means 3T3-L1 adipocytes were collected and scraped from the plates with lysis buffer (1% Triton X-100 in PBS) for follow-up experimental analysis.

  1. Astaxanthin was added to cell culture ten days after the differentiation process so the effect of astaxanthin was evaluated on mature adipocytes, not on adipogenesis!

RESPONSE: Thank you for your suggestion. The word “adipogenesis” has been corrected to “lipid accumulation” in the sections of title, abstract, keywords, result and discussion.

  1. There is no cytotoxicity-viability test of astaxanthin on adipocytes; maybe this lipid-lowering effect was due to cytotoxicity of astaxanthin?

RESPONSE: Thank you for your suggestion. However, we believe that oil red O stained material (OROSM) was used to estimate the cell number of adipocytes. In the present study, 3T3-L1 adipocytes were treated with 0-25 µg/mL astaxanthin for 24 and 48 hours have no cell cytotoxicity (Fig. 3B, cell number> 80%). Chintong et al., (2019) indicated that human dermal fibroblast cells were treated with a concentration of astaxanthin in the range of 5-160 µg/mL, no change of cell morphology. Obviously, the astaxanthin extract at all concentrations was non-toxic to the cells. The dose of astaxanthin used in our study was much lower. Thus, in this experiment it is speculated that astaxanthin has no cell cytotoxicity.

Reference

Chintong, S., Phatvej, W., Rerk-Am, U., Waiprib, Y., Klaypradit, W. (2019). In vitro antioxidant, antityrosinase, and cytotoxic activities of astaxanthin from shrimp waste. Antioxidants, 8(5), 128.

  1. The real-time PCR was normalized to GAPDH, not to b-actin, GAPDH expression 0-25 levels increased during the course of 3T3-L1 adipocyte differentiation and b-actin show relatively stable expression level [Zhang J, Tang H, Zhang Y, Deng R, Shao L, Liu Y, Li F, Wang X, Zhou L. Identification of suitable reference genes for quantitative RT-PCR during 3T3-L1 adipocyte differentiation. Int J Mol Med. 2014 May;33(5):1209-18. doi: 10.3892/ijmm.2014.1695]. In this study have been found that the mRNA expression levels of glyceraldehyde-3-phosphate dehydrogenase (GAPDH) and transferrin receptor (TFRC) significantly increased during 3T3-L1 adipocyte differentiation.

RESPONSE: Recent research shows that GAPDH was still used in 3T3-L1 adipocytes as the housekeeper gene (Li et al., 2016; Takeuchi et al., 2020). Therefore, we believe that the GAPDH gene as a housekeeping gene still has certain credibility.

References

  1. Takeuchi, H., Takahashi-Muto, C., Nagase, M., Kassai, M., Tanaka-Yachi, R., Kiyose, C. (2020). Anti-inflammatory Effects of Extracts of Sweet Basil (Ocimum basilicum L.) on a Co-culture of 3T3-L1 Adipocytes and RAW264. 7 Macrophages. Journal of Oleo Science, ess19321.
  2. Li, Y., Zhao, X., Feng, X., Liu, X., Deng, C., Hu, C. H. (2016). Berberine alleviates olanzapine-induced adipogenesis via the AMPKα–SREBP pathway in 3T3-L1 cells. International journal of molecular sciences, 17(11), 1865.

  1. In my opinion, this is a preliminary study which examined only one aspect – the effect of astaxanthin on lipid accumulation in mature adipocytes, which was examined by Oil Red and triglyceride assay kit and six gene expression, so the sentence in conclusions “These findings suggested that astaxanthin have a potent antioxidant carotenoid, as a promising natural candidate for prevented lifestyle diseases and health promotion” is overstated and more study is needed to be done to drawing such conclusions (like investigation on lipolysis, adipogenesis, leptin, adiponectin expression etc.)

RESPONSE: Thank you for your suggestion. We have redrafted the conclusion section (p. 0, lines 250-251) to establish a clearer focus. “These findings suggested that … may provide a potential therapeutic approach for the treatment of obesity.” instead of “These findings suggested… carotenoid, as a promising natural candidate for prevented lifestyle diseases and health promotion”

  1. Fig. 2, the photos of adipocytes are not sharp and not clear; it is impossible to see round cells filled with lipids droplets. Moreover, all photos should be taken in the same zoom that is possible to see adipocytes morphology, here adipocytes seem to be deformed; it looks like there is a cytotoxic effect of astaxanthin.

RESPONSE:

(1) Thank you for your suggestion. We have adjusted the sharp and clear of the photos of adipocytes (Fig. 1, original Fig. 2).

(2) In this manuscript, the morphological characteristics of lipid accumulation in cells were examined by microscopy at × 200 magnification

(3) The question about cytotoxicity, please see Q6 above.

Reviewer 2 Report

         This study suggested that astaxanthin efficiently suppressed adipogenesis in 3T3-L1 adipocytes and its action is associated with the down-regulation of adipogenesis-related genes and the triglyceride accumulation in 3T3-L1 adipocytes.

          Several evidence suggests that visceral adipose tissue is a metabolic and inflammatory organ that signals and modulates the action and metabolism of the brain, liver, muscle and cardiovascular system [1]. The progress of the hepatic steatosis (HS), a clinicopathological status, is influenced by cellular oxidative stress, lipogenesis, fatty acid (FA) oxidation, and inflammatory responses. Astaxanthin (ATX) is a highly potent antioxidant which can be extracted from Haematococcus pluvialis when cultivated and induced at high stress conditions [2]. ATX inhibite endoplasmic reticulum stress and lipogenesis at the intracellular level [3]. ATX is effective in inhibiting cell death, lipotoxicity, and inflammation [3].

           Authors are kindly requested to emphasize the current concepts about these issues in the context of recent knowledge and the available literature. This articles should be quoted in the References list.

References

  1. What is the role of adiponectin in obesity related non-alcoholic fatty liver disease?. World J Gastroenterol. 2013; 19 (6): 802‐ doi:10.3748/wjg.v19.i6.802.
  2. Enhanced proliferation and differentiation of mesenchymal stem cells by astaxanthin-encapsulated polymeric micelles. PLoS One. 2019;14(5):e0216755. Published 2019 May 20. doi:10.1371/journal.pone.0216755.
  3. Effects of Antioxidants in Reducing Accumulation of Fat in Hepatocyte. Int J Mol Sci. 2018; 19 (9): 2563. Published 2018 Aug 29. doi:10.3390/ijms19092563.

Author Response

Reviewer 2

Comments and Suggestions for Authors

         This study suggested that astaxanthin efficiently suppressed adipogenesis in 3T3-L1 adipocytes and its action is associated with the down-regulation of adipogenesis-related genes and the triglyceride accumulation in 3T3-L1 adipocytes.

         Several evidence suggests that visceral adipose tissue is a metabolic and inflammatory organ that signals and modulates the action and metabolism of the brain, liver, muscle and cardiovascular system [1]. The progress of the hepatic steatosis (HS), a clinicopathological status, is influenced by cellular oxidative stress, lipogenesis, fatty acid (FA) oxidation, and inflammatory responses. Astaxanthin (ATX) is a highly potent antioxidant which can be extracted from Haematococcus pluvialis when cultivated and induced at high stress conditions [2]. ATX inhibit endoplasmic reticulum stress and lipogenesis at the intracellular level [3]. ATX is effective in inhibiting cell death, lipotoxicity, and inflammation [3].

           Authors are kindly requested to emphasize the current concepts about these issues in the context of recent knowledge and the available literature. This articles should be quoted in the References list.

References

What is the role of adiponectin in obesity related non-alcoholic fatty liver disease?. World J Gastroenterol. 2013; 19 (6): 802‐ doi:10.3748/wjg.v19.i6.802.

Enhanced proliferation and differentiation of mesenchymal stem cells by astaxanthin-encapsulated polymeric micelles. PLoS One. 2019;14(5):e0216755. Published 2019 May 20. doi:10.1371/journal.pone.0216755.

Effects of Antioxidants in Reducing Accumulation of Fat in Hepatocyte. Int J Mol Sci. 2018; 19 (9): 2563. Published 2018 Aug 29. doi:10.3390/ijms19092563.

RESPONSE: We agree with your assessment. We have re-written this part, according to the Reviewer’s suggestion to establish a clearer focus. (p.1-2, lines 43-46; p.2, lines 56-58; p. 2, lines 62-63).

Reviewer 3 Report

Tsai et al., reported the effect of astaxanthin on adipogenesis inhibition in 3T3-L1 adipocytes.

Oil red O stained material was found to be only signiciantly reduced at 25 μg/ml of astaxanthin treatment on 3T3-L1 adipocytes for 24 and 48 hours. But at lower concentration of astaxanthin, oil red O stained material was not decreased. It was even significantly increased at 10 μg/ml of astaxanthin treatment. This result must be explained and discussed.

For Figure 4B, why the intracellular triglyceride content is lower after treatment of lower concentration of astaxanthin (5 μg/ml) than higher concentrations (10 μg/ml and 25 μg/ml)? But no differences on GPDH activity among these three concentrations were observed (Figure 5). This result must be explained and discussed.

Further qRT-PCR have been performed to analyze gene expression of peroxisome proliferator-activated receptor-γ (PPARγ) and its targets. Most genes were downregulated after longer treatmet (6-9 days).

According to all the results obtained in this study, it is hard to see a trend. For example, is the effect of astaxanthin is dose-dependent? What are you expecting for even higher concentration of astaxanthin (>25 μg/ml)? What is the most effective concentration? This study did not have enough points of astaxanthin concentration? This must be discussed in Discussion?  

Also, are there any other compounds that have been reported that can inhibit adipogenesis in 3T3-L1 adipocytes? This points should be mentioned in Introduction. Furthermore, What are the advantages and disadvantages of astaxanthin compared to these compounds? This should be discussed in Discussion. Morever, this is important to recongnize the novety of this study.

Some other specific comments:   

All concentrations should be labelled in each Fig A.

Line 22, delete “that”.

Line 40, only one citation should be provided following the journal instruction for the whole text.

Line 43, full name of GPDH must be provided.

Line 52-53, this sentence should be rewrite.

Line 57, “Although” is unproperly used here.

Line 67, what is 3T3-L1 adipocytes? Why is it special? Why you want to use it but not other cells in the study?

Line 73-75, this sentence must be rewritten.

Line 88, “The data revealed that treated with …”, this sentence should be rewritten.

Line 101, delete “resulted in”.

Lines 102-103, this sentence should be revised.

Lines 111-112, this sentence must be rewritten.

Lines 123-124, this sentence must be rewritten.

Line 126, a period is missing.

Line 145, revise “In gene expressions” to “For gene expression”.

Line 168, if this is the fourth section, all the figure numbers need to be changed. The first figure appered in the manuscript should be numbered as Figure 1.

Figure 1, the size of all figs and table should be changes properly, all the words and numbers are not easy to read.

In summary, major revision is needed.

Author Response

Reviewer 3

Comments and Suggestions for Authors

Tsai et al., reported the effect of astaxanthin on adipogenesis inhibition in 3T3-L1 adipocytes.

Oil red O stained material was found to be only signiciantly reduced at 25 μg/ml of astaxanthin treatment on 3T3-L1 adipocytes for 24 and 48 hours. But at lower concentration of astaxanthin, oil red O stained material was not decreased. It was even significantly increased at 10 μg/ml of astaxanthin treatment. This result must be explained and discussed.

RESPONSE: We agree with your assessment. However, our research data revealed that treated with 5 μg/mL of astaxanthin significantly decreased the level of OROSM at 24 and 48 hours in 3T3-L1 adipocytes (98.25 and 99.85%, respectively). Moreover, treated with 10 μg/mL of astaxanthin significantly decreased the level of OROSM at 24 and 48 hours in 3T3-L1 adipocytes (from 100% to 98.25%). The above results showed that OROSM values have little change.

For Figure 4B, why the intracellular triglyceride content is lower after treatment of lower concentration of astaxanthin (5 μg/ml) than higher concentrations (10 μg/ml and 25 μg/ml)? But no differences on GPDH activity among these three concentrations were observed (Figure 5). This result must be explained and discussed.

RESPONSE: This is an interesting perspective. We have rewritten the discussion (p. 7, lines 146-150). “Previously reports have proposed that several strategies can be used to improve lipid metabolism [21]. Price et al. (2012) [24] indicated that the differences signaling pathways resveratrol treatment can have at different doses. Therefore, astaxanthin could be a valuable component of improving obesity in optimal doses (5 µg/mL)”

Reference

Price, N.L,; Gomes, A.P,; Ling, A.J,; Duarte, F.V,; Martin-Montalvo, A,; North, B.J,; Agarwal, B,; Ye, L,; Ramadori, G,; Teodoro, J.S,; Hubbard, B.P,; Varela, A.T,; Davis, J.G,; Varamini, B,; Hafner, A,; Moaddel, R,; Rolo, A.P,; Co;ppari, R,; Palmeira, C;M,; de Cabo, R,; Baur, J.A,; Sinclair, D.A. SIRT1 is required for AMPK activation and the beneficial effects of resveratrol on mitochondrial function. Cell Metab. 2012, 15, 675-690. doi: 10.1016/j.cmet.2012.04.003.

Further qRT-PCR have been performed to analyze gene expression of peroxisome proliferator-activated receptor-γ (PPARγ) and its targets. Most genes were downregulated after longer treatmet (6-9 days).

RESPONSE: Thanks for your comments.

According to all the results obtained in this study, it is hard to see a trend. For example, is the effect of astaxanthin is dose-dependent? What are you expecting for even higher concentration of astaxanthin (>25 μg/ml)? What is the most effective concentration? This study did not have enough points of astaxanthin concentration? This must be discussed in Discussion?

RESPONSE:

Thank you for your comments. From the results of cell experiments, it can be seen that astaxanthin inhibits lipid accumulation in 3T3-L1 adipocytes. In the future work, we will conduct a study on the effect of astaxanthin on obese rats induced by a high-fat diet, and hope to publish the results in high-quality journals, such as molecules.

Also, are there any other compounds that have been reported that can inhibit adipogenesis in 3T3-L1 adipocytes? This points should be mentioned in Introduction. Furthermore, What are the advantages and disadvantages of astaxanthin compared to these compounds? This should be discussed in Discussion. Morever, this is important to recongnize the novety of this study.

RESPONSE:

Our research team previously published many studies on phytochemicals have inhibits lipid accumulation and inhibits high fat diet-induced obese rats. We believe that the advantages of astaxanthin are effective at relatively low effective dose concentrations.

Chang, W. T., Wu, C. H., Hsu, C. L.* Diallyl trisulphide inhibits adipogenesis in 3T3-L1 adipocytes through lipogenesis, fatty acid transport, and fatty acid oxidation pathways. Journal of Functional Foods, 2015, 16, 414-422

Hsu, C. L., Lin, Y. J., Ho, C. T., Yen, G. C. The inhibitory effect of pterostilbene on inflammatory responses during the interaction of 3T3-L1 adipocytes and RAW 264.7 macrophages. Journal of Agricultural and Food Chemistry, 2013, 61 (3), 602-610.

Hsu, C. L., Lin, Y. J., Ho, C. T., Yen, G. C. Inhibitory effects of garcinol and pterostilbene on cell proliferation and adipogenesis in 3T3-L1 cells. Food & Function, 2012, 3 (1), 49-57.

Yen, G. C., Chen, Y. C., Chang, W. T., Hsu, C. L.* Effects of polyphenolic compounds on tumor necrosis factor-a (TNF-a)-induced changes of adipokines and oxidative stress in 3T3-L1 adipocytes. Journal of Agricultural and Food Chemistry, 2011, 59 (2): 546-551.

Hsu, C. L., Yen, G. C. Effects of flavonoids and phenolic acids on the inhibition of adipogenesis in 3T3-L1 adipocytes. Journal of Agricultural and Food Chemistry, 2007, 55 (21): 8404-8410.

Hsu, C. L., Lo, W. H., Yen, G. C. Gallic acid induces apoptosis in 3T3-L1 pre-adipocytes via a Fas- and mitochondria-mediated pathway. Journal of Agricultural and Food Chemistry, 2007, 55 (18): 7359-7365.

Hsu, C. L., Yen, G. C. Effects of capsaicin on induction of apoptosis and inhibition of adipogenesis in 3T3-L1 cells. Journal of Agricultural and Food Chemistry, 2007, 55 (5): 1730-1736.

Hsu, C. L., Yen, G. C. Induction of cell apoptosis in 3T3-L1 pre-adipocytes by flavonoids is associated with their antioxidant activity. Molecular Nutrition & Food Research, 2006, 50 (11): 1072-1079.

Hsu, C. L., Huang, S. L., Yen, G. C. Inhibitory effect of phenolic acids on proliferation of 3T3-L1 preadipocytes in relation to their antioxidant activity. Journal of Agricultural and Food Chemistry, 2006, 54 (12): 4191-4197.

Some other specific comments:

All concentrations should be labelled in each Fig A.

RESPONSE:

Thank you for your comments. We have revised your opinion.

Line 22, delete “that”.

RESPONSE:

Thank you for your comments. The word “that” has been deleted.

Line 40, only one citation should be provided following the journal instruction for the whole text.

RESPONSE:

Thank you for your comments. We have revised your opinion.

Line 43, full name of GPDH must be provided.

RESPONSE:

Thank you for your comments. The full name has been added.

Line 52-53, this sentence should be rewrite.

RESPONSE:

Thank you for your comments. The sentence has been rewritten.

Line 57, “Although” is unproperly used here.

RESPONSE:

Thank you for your comments. We have revised your opinion.

Line 67, what is 3T3-L1 adipocytes? Why is it special? Why you want to use it but not other cells in the study?

RESPONSE:

Thank you for your comments. We have revised your opinion.

Line 73-75, this sentence must be rewritten.

RESPONSE:

Thank you for your comments. The sentence has been rewritten.

Line 88, “The data revealed that treated with …”, this sentence should be rewritten.

RESPONSE:

Thank you for your comments. The sentence has been rewritten.

Line 101, delete “resulted in”.

RESPONSE:

Thank you for your comments. The word “resulted in” has been deleted.

Lines 102-103, this sentence should be revised.

RESPONSE:

Thank you for your comments. The sentence has been rewritten.

Lines 111-112, this sentence must be rewritten.

RESPONSE:

Thank you for your comments. The sentence has been rewritten.

Lines 123-124, this sentence must be rewritten.

RESPONSE:

Thank you for your comments. The sentence has been rewritten.

Line 126, a period is missing.

RESPONSE:

Thank you for your comments. We have revised your opinion.

Line 145, revise “In gene expressions” to “For gene expression”.

RESPONSE:

Thank you for your comments. We have revised your opinion.

Line 168, if this is the fourth section, all the figure numbers need to be changed. The first figure appered in the manuscript should be numbered as Figure 1.

RESPONSE:

Thank you for your comments. We have revised your opinion.

Figure 1, the size of all figs and table should be changes properly, all the words and numbers are not easy to read.

RESPONSE:

Thank you for your comments. We have revised your opinion.

Round 2

Reviewer 1 Report

The authors of the manuscript didn't improve the first version of the document, only changed a few sentences. This scientific paper is based on only 3 methods (Oil Red o and triglyceride accumulation measure the same - the amount of lipids in cells, so shows the same results), next simple method is GPDH activity and the third method is real-time PCR of 6 genes. In my opinion, for the scientific paper, it is not enough, in such form is a basic-preliminary study. Besides still lack of information about cytotoxicity of astaxanthin - Oil red O only dye lipids, don't show a number of live and dead cells!!! Still, the description of methods is very unclear, why cells were scrapped and then treated with astaxanthin, this is illogical, first cells should be treated then harvested, so it leads me to doubt that experiments were performed appropriately.

Author Response

Reviewer 1

Comments and Suggestions for Authors

The authors of the manuscript didn't improve the first version of the document, only changed a few sentences. This scientific paper is based on only 3 methods (Oil Red o and triglyceride accumulation measure the same - the amount of lipids in cells, so shows the same results), next simple method is GPDH activity and the third method is real-time PCR of 6 genes. In my opinion, for the scientific paper, it is not enough, in such form is a basic-preliminary study. Besides still lack of information about cytotoxicity of astaxanthin - Oil red O only dye lipids, don't show a number of live and dead cells!!! Still, the description of methods is very unclear, why cells were scrapped and then treated with astaxanthin, this is illogical, first cells should be treated then harvested, so it leads me to doubt that experiments were performed appropriately.

Reply: Thank you for your valuable comments. Maybe our previous reply is not clear, which makes you have doubts about the 3T3-L1 adipocytes test. We have published more than 9 SCI journal papers on the subject of 3T3-L1 pre-adipocytes and adipocytes. The in vitro 3T3-L1 adipocyte test as a screening model for evaluation of anti-obesity action in my research. The future research will focus on the effects of astaxanthin in high-fat diet induced obese rats and its molecular mechanism. It is true that oil red staining cannot distinguish the survival and death of 3T3-L1 adipocytes. However, the OROSM was expressed relative to the

number of cells counted on comparable plates. McNeel et al. (2003) indicated that have used the results of oil red staining as a cell number assessment tool. 3T3-L1 adipocytes after ten days of differentiation and culture to become mature 3T3-L1 adipocytes. 3T3-L1 adipocytes were incubated with 0, 1, 5, 10, and 25 µg/mL of astaxanthin for 24 and 48 hours at 37°C in a humidified 5% CO2 incubator. The above description, I hope to enhance your core value of the article content.

McNeel, R. L.; Mersmann, H. J. Effects of isomers of conjugated linoleic acid on porcine adipocyte growth and differentiation. J. Nutr. Biochem. 2003, 14, 266-274.

Our published SCI journal papers:

Chang, W. T., Wu, C. H., Hsu, C. L.* Diallyl trisulphide inhibits adipogenesis in 3T3-L1 adipocytes through lipogenesis, fatty acid transport, and fatty acid oxidation pathways. J. Funct. Foods, 2015, 16, 414-422

Hsu, C. L., Lin, Y. J., Ho, C. T., Yen, G. C. The inhibitory effect of pterostilbene on inflammatory responses during the interaction of 3T3-L1 adipocytes and RAW 264.7 macrophages. J. Agric. Food Chem., 2013, 61 (3), 602-610.

Hsu, C. L., Lin, Y. J., Ho, C. T., Yen, G. C. Inhibitory effects of garcinol and pterostilbene on cell proliferation and adipogenesis in 3T3-L1 cells. Food Funct., 2012, 3 (1), 49-57.

Yen, G. C., Chen, Y. C., Chang, W. T., Hsu, C. L.* Effects of polyphenolic compounds on tumor necrosis factor-a (TNF-a)-induced changes of adipokines and oxidative stress in 3T3-L1 adipocytes. J. Agric. Food Chem., 2011, 59 (2): 546-551.

Hsu, C. L., Yen, G. C. Effects of flavonoids and phenolic acids on the inhibition of adipogenesis in 3T3-L1 adipocytes. J. Agric. Food Chem., 2007, 55 (21): 8404-8410.

Hsu, C. L., Lo, W. H., Yen, G. C. Gallic acid induces apoptosis in 3T3-L1 pre-adipocytes via a Fas- and mitochondria-mediated pathway. J. Agric. Food Chem., 2007, 55 (18): 7359-7365.

Hsu, C. L., Yen, G. C. Effects of capsaicin on induction of apoptosis and inhibition of adipogenesis in 3T3-L1 cells. J. Agric. Food Chem., 2007, 55 (5): 1730-1736.

Hsu, C. L., Yen, G. C. Induction of cell apoptosis in 3T3-L1 pre-adipocytes by flavonoids is associated with their antioxidant activity. Mol. Nutr. Food Res., 2006, 50 (11): 1072-1079.

Hsu, C. L., Huang, S. L., Yen, G. C. Inhibitory effect of phenolic acids on proliferation of 3T3-L1 preadipocytes in relation to their antioxidant activity. J. Agric. Food Chem., 2006, 54 (12): 4191-4197.

Reviewer 3 Report

Tsai et al., replied to all of my comments. Please see below some of my further comments based on their responses.  

  1. For my first comment, the authors replied that “Moreover, treated with 10 μg/mL of astaxanthin significantly decreased the level of OROSM at 24 and 48 hours in 3T3-L1 adipocytes (from 100% to 98.25%).” But this is different from what was shown in Fig. 2A and Fig. 2B. I assume that the OROSM at 0 μg/ml is 100%, the bars at 10 μg/ml in both 2A and Fig. 2B are apparently higher than that at 0 μg/ml. How can this be possible? Also, at both 24 and 48 hours with 10 μg/mL of astaxanthin, the level of OROSM decreased from 100% to 98.25%?

  1. For my first comment, In the legend of Fig. 2, the authors stated that mean values with different letters were significantly different (p < 0.05), it indicated that the difference for the level of OROSM between 0 μg/ml (shown as b) and 10 μg/ml (shown as a) treatment of astaxanthin is significant. But the authors in the response indicated that “the above results showed that OROSM values have little change.” Then at least this should be mentioned in the result part.   

  1. Also, are there any other studies on phytochemicals that inhibit lipid accumulation that not besides the Hsu lab? The authors believe that the advantages of astaxanthin are effective at relatively low effective dose concentrations. I suggest the authors include this point in the Discussion by comparing some of the published phytochemicals (not just from the same lab) with effective dose concentrations.

Author Response

Reviewer 3

Tsai et al., replied to all of my comments. Please see below some of my further comments based on their responses.

1. For my first comment, the authors replied that “Moreover, treated with 10  μg/mL of astaxanthin significantly decreased the level of OROSM at 24 and 48 hours in 3T3-L1 adipocytes (from 100% to 98.25%).” But this is different from what was shown in Fig. 2A and Fig. 2B. I assume that the OROSM at 0 μg/ml is 100%, the bars at 10 μg/ml in both 2A and Fig. 2B are apparently higher than that at 0 μg/ml. How can this be possible? Also, at both 24 and 48 hours with 10 μg/mL of astaxanthin, the level of OROSM decreased from 100% to 98.25%?

Reply: Thank you for your suggestion. We have rechecked the data of oil red O stained material (OROSM) and confirm the statistical analysis. According to statistical analysis, we have revised the graph of Figure 2. The results showed that the astaxanthin (treated with 1-10 µg/mL for 24 and 48 hours) were no significant differences compared to 0 µg/mL. The treatment of 3T3-L1 adipocytes with astaxanthin (25 µg/mL, 24h and 48h) were significantly decreased adipocytes numbers from 100% to 94.56% (24 hours) and 86.21% (48 hours), respectively (p <0.05). So, the sentence “Moreover, treated with 1-10 µg/mL of astaxanthin were showed no significantly reduced the level of OROSM at 24 and 48 hours in 3T3-L1 adipocytes.” have been added. (Please see page 2, lines 77-79)

2. For my first comment, In the legend of Fig. 2, the authors stated that mean  values with different letters were significantly different (p < 0.05), it indicated that the difference for the level of OROSM between 0 μg/ml (shown as b) and 10 μg/ml (shown as a) treatment of astaxanthin is significant. But the authors in the response indicated that “the above results showed that OROSM values have little change.” Then at least this should be mentioned in the result part.

Reply: Thank you for your suggestion. The sentence “Moreover, treated with 1-10 µg/mL of astaxanthin were showed no significantly reduced the level of OROSM at 24 and 48 hours in 3T3-L1 adipocytes.” have been added. (Please see page 2, lines 77-79)

3. Also, are there any other studies on phytochemicals that inhibit lipid accumulation that not besides the Hsu lab? The authors believe that the advantages of astaxanthin are effective at relatively low effective dose concentrations. I suggest the authors include this point in the Discussion by comparing some of the published phytochemicals (not just from the same lab) with effective dose concentrations.

Reply:Thank you for your suggestion. Some references have been added.

The sentences “Astaxanthin is the most common carotenoid in marine organisms, such as algae and aquatic animals. Previous studies indicated that astaxanthin has multiple pharmacological properties including antioxidant, anti-inflammatory, anti-cancer activities, and anti-adiposity action (Choi et al., 2019).” have been added. (Please see page 7, lines 137-139).

The sentences “Inoue et al. [18] demonstrated that the astaxanthin (10 mM) significantly reduced lipid accumulation of 3T3-L1 adipocytes by specifically inhibiting PPARg transcriptional activity. Jia et al. [24] also showed that astaxanthin has a significantly reduced intracellular triglyceride and cholesterol when HepG2 cells were treated with 5 and 10 mM astaxanthin for 24 hours.” have been added. (Please see page 7, lines 146-150).

The sentences “Zhao et al. (2017) reported that carotenoids (such as bixin, lycopene, and β-carotene) significantly decreased lipid accumulation in 3T3-L1 adipocytes and markedly downregulate the protein levels of PPARg, FABP4, leptin, and ACC.” have been added. (Please see page 7, lines 161-164).

The references have been added. (Please see page 11, lines 326-333 and 345-346)

22. Choi, C.I. Astaxanthin as a peroxisome proliferator-activated receptor (PPAR) modulator: Its therapeutic implications. Mar. Drugs 2019, 17, 242. PMID: 31018521

24. Jia, Y.; Kim, J.Y.; Jun, H.J.; Kim, S.J.; Lee, J.H.; Hoang, M.H.; Hwang, K.Y.; Um, S.J.; Chang, H.I.; Lee, S.J. The natural carotenoid astaxanthin, a PPAR-a agonist and PPAR-g antagonist, reduces hepatic lipid accumulation by rewiring the transcriptome in lipid-loaded hepatocytes. Mol. Nutr. Food Res. 2012, 56, 878–888.

28. Zhao, W.E.; Fan, J.; Gao, R.; Ngoc, N.B. Suppressive effects of carotenoids on proliferation and differentiation of 3T3-L1 Preadipocytes. Anim. Sci. J. 2017, 5, 129-136.
